# Early Postoperative Immunothrombosis of Bioprosthetic Mitral Valve and Left Atrium: A Case Report

**DOI:** 10.3390/ijms23126736

**Published:** 2022-06-16

**Authors:** Alexander Kostyunin, Tatiana Glushkova, Alexander Stasev, Rinat Mukhamadiyarov, Elena Velikanova, Leo Bogdanov, Anna Sinitskaya, Maxim Asanov, Evgeny Ovcharenko, Leonid Barbarash, Anton Kutikhin

**Affiliations:** Department of Experimental Medicine, Research Institute for Complex Issues of Cardiovascular Diseases, 650002 Kemerovo, Russia; kostae@kemcardio.ru (A.K.); glushtv@kemcardio.ru (T.G.); stasan@kemcardio.ru (A.S.); mukhra@kemcardio.ru (R.M.); veliea@kemcardio.ru (E.V.); bogdla@kemcardio.ru (L.B.); cepoav@kemcardio.ru (A.S.); asanma@kemcardio.ru (M.A.); ovchea@kemcardio.ru (E.O.); barbls@kemcardio.ru (L.B.)

**Keywords:** bioprosthetic heart valves, recurrent thrombosis, prosthetic valve thrombosis, left atrial thrombosis, atrial fibrillation, innate immune response, neutrophils, neutrophil extracellular traps, immunothrombosis, anticoagulant therapy

## Abstract

A 72-year-old female patient with mixed rheumatic mitral valve disease and persistent atrial fibrillation underwent mitral valve replacement and suffered from a combined thrombosis of the bioprosthetic valve and the left atrium as soon as 2 days post operation. The patient immediately underwent repeated valve replacement and left atrial thrombectomy. Yet, four days later the patient died due to the recurrent prosthetic valve and left atrial thrombosis which both resulted in an extremely low cardiac output. In this patient’s case, the thrombosis was notable for the resistance to anticoagulant therapy as well as for aggressive neutrophil infiltration and release of neutrophil extracellular traps (NETs) within the clot, as demonstrated by immunostaining. The reasons behind these phenomena remained unclear, as no signs of sepsis or contamination of the BHV were documented, although the patient was diagnosed with inherited thrombophilia that could impede the fibrinolysis. The described case highlights the hazard of immunothrombosis upon valve replacement and elucidates its mechanisms in this surgical setting.

## 1. Introduction

Bioprosthetic heart valves (BHVs) are commonly employed in valve replacement surgery, to avoid the complications associated with a lifelong use of anticoagulants upon the implantation of a mechanical valve [1]. For decades, BHV thrombosis was considered as a rare phenomenon, yet recent studies have shown that it is responsible for ≈12% cases of BHV failure [2]. The recent estimates indicated the annual rate of BHV thrombosis as around 1.5% [2]. The implementation of minimally invasive transcatheter heart valve replacement with a higher risk of thromboembolism, as compared to surgical valve replacement, underscores the importance of proper antithrombotic therapy [3]. Albeit the causes and pathogenesis of BHV thrombosis are still poorly understood, patients with hypercoagulability, venous thromboembolism, atrial fibrillation, and severe left ventricular dysfunction belong to a high-risk group [4,5]. An association between BHV thrombosis and inflammatory syndromes, including eosinophilia [6], allergic acute coronary (Kounis) syndrome [7], and antiphospholipid syndrome [8] has also been reported. Here, we describe a case of recurrent bioprosthetic mitral valve and left atrial thrombosis which occurred in the early postoperative period in a 72-year-old female patient with mixed rheumatic mitral valve disease and persistent atrial fibrillation. In this case, the thrombosis was accompanied by a neutrophil invasion and activation of NETosis within the clot.

## 2. Case Presentation

A 72-year-old female patient was admitted to the Department of Cardiovascular Surgery in the Research Institute of Complex Issues of Cardiovascular Diseases in January 2019 with dyspnea on exertion, but rarely at rest. The past surgical history included appendectomy in 1954, endoscopic cholecystectomy in 2017, and femoral neck osteosynthesis in 2017. The patient was also diagnosed with Gilbert’s syndrome, arterial hypertension (maximum blood pressure 170/100 mmHg, daily blood pressure 130/80 mmHg), but did not have acute infectious diseases, diabetes mellitus, or dyslipidemia, nor a past medical history of myocardial infarction, cerebrovascular events, or blood transfusion. The patient was earlier prescribed warfarin, perindopril, atorvastatin, spironolactone and torsemide.

The echocardiography and cardiac ventriculography revealed mixed rheumatic mitral valve disease (effective orifice area 1.2 cm^2^, grade III-IV regurgitation), tricuspid valve regurgitation (III-IV grade), persistent atrial fibrillation, left ventricular hypertrophy, coronary artery disease, and chronic heart failure (NYHA class II). Multi-slice computed tomography indicated a moderate-to-severe calcification of the aortic arch and coronary arteries. The patient was assigned to mitral valve replacement, correction of the tricuspid valve regurgitation, and coronary artery bypass graft surgery. As expected in Gilbert’s syndrome, the blood analysis showed hyperbilirubinemia (≈64 mmol/L), yet the hepatoprotective drugs prescribed before the surgery lowered the serum bilirubin to 43.9 mmol/L. The coagulation parameters were as follows (Table 1): international normalized ratio 1.2; prothrombin time 69% (Quick-type); fibrinogen 5.2 g/L; activated partial thromboplastin time 29 s; protein C activity 80%; antithrombin III 86.3%; XIIa-dependent fibrinolysis 14 min. The level of the soluble fibrin monomer complexes (17.0 mg per 100 mL) was around four-fold higher than th maximum reference value (<4.0 mg per 100 mL). The complete blood count analysis showed that platelets, red blood cells, erythrocyte sedimentation rate, hematocrit, and white blood cells were within or near the reference range (Table 2). Regarding the pre-operative anticoagulant therapy, the patient was prescribed warfarin, which was replaced with heparin 5 days before the surgery, in accordance with the respective guidelines [9].

On 30 January, the patient underwent a mitral valve replacement using UniLine bioprosthetic atrioventricular valve of size 30 (NeoCor, Kemerovo, Russia) [10], tricuspid valve annuloplasty using a respective annuloplasty ring of size 32 (NeoCor, Kemerovo, Russia), and coronary artery bypass graft surgery. The surgical procedure was not accompanied by emergent complications. During the surgery, we applied a standard algorithm of heparin treatment during the cardiopulmonary bypass.

In the early postoperative period, high doses of inotropes and vasopressors were required. The repeated echocardiography revealed a reduction in the left ventricular ejection fraction to 16%, an increase in the end-diastolic volume from 90 to 160 mL, and zones of hypo/akinesis. The postoperative coronary angiography confirmed the primary patency of coronary arteries and the graft. On the morning of 31 January, we made a biventricular bypass with an extracorporeal membrane oxygenation, keeping the activated clotting time at 180 s at heparin infusion. However, the cardiac output was still poor. The coagulation parameters were as follows (Table 1): international normalized ratio 1.25; prothrombin time 63% (Quick-type); soluble fibrin monomer complexes 6.5 mg per 100 mL; activated partial thromboplastin time 53 s. The platelet function testing showed a decreased platelet aggregation at adrenaline, whilst the other parameters were within the reference range. A complete blood count indicated augmented erythrocyte sedimentation rate (24 mm/h) and increased levels of white blood cells (14 × 10^9^/L), and segmented neutrophils (92%), as well as a reduced platelet count (94 × 10^9^/L) (Table 2).

The next day (1 February), the echocardiography identified a prosthetic valve and left atrial thrombosis. The patient was immediately transferred to the operating room for a left atrial thrombectomy and for a repeated valve replacement using a similar bio prosthesis model. Again, we adhered to the standard protocol of heparin treatment at cardiopulmonary bypass. During the following 4 days, the therapy remained the same. The coagulation parameters were as follows (Table 1): international normalized ratio 1.18–1.33; prothrombin time 54–71% (Quick-type); soluble fibrin monomer complexes 4.5–10.0 mg per 100 mL; activated partial thromboplastin time 50–115 s. The complete blood count was characterized by a notable increase in the erythrocyte sedimentation rate (51–59 mm/h) and thrombocytopenia (15–69 platelets × 10^9^/L) (Table 2). Urine inoculation detected *Escherichia coli*, yet inoculation and specific PCR tests of blood, the excised native and prosthetic valves, and the left atrial thrombus were negative. The GPIIb/IIIa inhibitors or fibrinolytic drugs were not employed because of the low platelet count and increased activated partial thromboplastin time, together indicating a high risk of bleeding in the context of the combined heart valve replacement and coronary artery bypass graft surgery. The patient did not have any signs or symptoms of autoimmune disorders; the rapid development of heart failure did not allow for the conduct of a comprehensive assessment of the respective biochemical markers.

On 5 February, the left ventricular ejection fraction reduced to 4–7%. Homogeneous masses with increased echogenicity in the left atrium testified to the recurrent left atrial thrombosis. The patient’s cardiac output was minimal and was supported by extracorporeal membrane oxygenation with a flow rate of 3.9 L/min. The trans-mitral flow was not observed. During the replacement of the oxygenator, the patient’s cardiac output remained inadequate which resulted in a cardiac arrest. Cardiopulmonary resuscitation was unsuccessful, which resulted in the death of the patient.

## 3. Investigation and Discussion

Rheumatic mitral stenosis is often complicated by atrial fibrillation [11]. Both of these disorders slow the blood flow and promote thrombosis through the activation of pro-coagulant and pro-inflammatory responses, eventually contributing to left atrial or valve thrombosis [12,13]. The left atrial thrombosis is frequently refractory to anticoagulant therapy [14]. Thus, the patient, diagnosed with mixed rheumatic mitral valve disease and persistent atrial fibrillation, was initially prone to thrombosis. To further uncover the mechanisms which led to the adverse outcome, we have interrogated the BHV excised after the primary mitral valve replacement; the second prosthetic valve was unavailable for research purposes due to the lack of consent for its excision from the relatives of the deceased patient.

A gross examination of the inflow side found a massive thrombus along the coaptation line (Figure 1A). The outflow side was free from thrombotic masses (Figure 1B). The excised BHV had no visible defects. Histological examination demonstrated a massive invasion of polymorphonuclear leukocytes to the thrombus. Although the immune cells were detected across the entire clot, the majority of them were found at the border between the thrombus and the leaflet surfaces made of bovine pericardium (Figure 1C). The polymorphonuclear leukocytes were also randomly distributed over the left atrial thrombotic masses (Figure 1D).

The electron microscopy confirmed the absence of immune cells inside the prosthetic valve (Figure 2A) and the neutrophil appearance of the immune cells in both the valvular clot (Figure 2A) and left atrial thrombus (Figure 2B).

To better investigate the cell populations within the thrombotic masses, we conducted immunohistochemical staining for the platelet marker CD41 (ab134131, Abcam), pan-leukocyte marker CD45 (ab10558, Abcam), macrophage marker CD68 (ab955, Abcam), pan-T cell marker CD3 (ab16669, Abcam), B lymphocyte marker CD19 (MA5-32544, Invitrogen), neutrophil elastase (ELA2, MAB91671-100, Novus Biologicals) and another neutrophil marker, myeloperoxidase (MPO, ab208670, Abcam). In addition, the clot was stained for fibrin (NBP2-50407, Novus Biologicals). Immune cells found within the prosthetic thrombus inconsistently expressed CD45, yet displayed abundant expression of ELA2 and MPO (Figure 3A). In conjunction with the electron microscopy findings, it confirmed the neutrophil identity of the immune cells within the clot. Moreover, immunofluorescence staining using the antibodies to ELA2 and citrullinated histone H3 (citH3, NB100-57135, Novus Biologicals), in combination with the chromatin staining (4′,6-diamidino-2-phenylindole, DAPI), detected neutrophil extracellular traps (NETs) in the prosthetic thrombus (Figure 3B).

Similar to the bioprosthetic clot, the immunohistochemical and immunofluorescence profiling confirmed the aggressive neutrophil infiltration (Figure 4A) and the presence of NETs (Figure 4B) in the left atrial thrombus.

We then assessed whether the patient had a genetic predisposition to thrombosis (i.e., thrombophilia, also called hypercoagulability) which ultimately led to the abnormal blood coagulation (Table 3). Genotyping took place for nine major gene polymorphisms associated with thrombophilia (rs1799963 (*F2* gene), rs6025 and rs6027 (*F5* gene), rs6046 (*F7* gene), rs5985 (*F13A1* gene), rs1800790 (*FGB* gene), rs1126643 (*ITGA2* gene), rs5918 (*ITGB3* gene), and the rs1799889 (*SERPINE1* gene)) and found minor alleles within four of them (G/T genotype for rs5985 polymorphism within the *F13A1* gene, C/T genotype for rs1126643 polymorphism within the *ITGA2* gene, T/C genotype for rs5918 polymorphism within the *ITGB3* gene, and 4G/4G genotype within the rs1799889 polymorphism within the *SERPINE1* gene). Collectively, these minor alleles impacted on the increased platelet aggregation/adhesion and compromised fibrinolysis (Table 3), which likely contributed to the recurrent prosthetic valve and left atrial thrombosis.

Neutrophils, the most abundant leukocyte lineage, are the effector cells of the innate immune system struggling with infectious agents and contributing to arterial and venous thrombosis, particularly in patients with sepsis and cancer [20,21]. Earlier, the neutrophils were shown to be the first responders to the implantation of medical devices [22,23], and studies in mouse models demonstrated that neutrophil adhesion and platelet deposition concurrently initiate thrombosis in the injured arteries and arterioles [24,25], as well as deep veins [26]. Moreover, the platelet depletion does not prevent neutrophil accumulation at the injury site [25], while the neutrophil depletion diminishes fibrin generation and hinders thrombosis [24,25]. These findings strongly suggested that the neutrophils are indispensable for the coagulation onset [27].

NETosis, a regulated form of neutrophil cell death, is notable for the chromatin unwinding, which results in the formation of neutrophil extracellular traps (NETs), the leading mechanism by which neutrophils induce blood coagulation [28]. The NETs tether erythrocytes, platelets, fibrin, and clotting factors [28]. The NETs also exhibit pro-coagulant properties, inducing thrombin generation and the activation and aggregation of platelets [28]. The accumulation of neutrophils and NETs in the blood clots was found in numerous thrombosis settings, including ischemic stroke [29,30], coronary artery thrombosis [31], and stent thrombosis [32]. The presence of NETs and aggressive neutrophil infiltration of the clot devoid of other immune cells suggested immunothrombosis as the mechanism behind the prosthetic valve and left atrial thrombosis in our patient. In combination with an inherited thrombophilia, it at least partially explains the inefficiency of the anticoagulant therapy and recurrent thrombosis, because the interactions between the DNA and histones of the uncoiled chromatin contributed to the mechanical strength and prolonged lysis of the fibrin clot, partially through binding of large fibrin degradation products [33]. Experiments demonstrated that the combination of DNase with tissue plasminogen activator (albeit it is inhibited by PAI-1, which was upregulated in our patient) might effectively promote clot lysis in an immunothrombosis scenario [33].

In contrast to the recurrent thrombosis, the causes of which are, to a certain extent, elucidated by genetic background, the reason for the neutrophil attack of the prosthetic valve remained unclear. The patient did not have sepsis, bacterial or mycotic contamination of the BHV, that was confirmed by negative blood inoculation, Gram staining of the bioprosthetic tissues (Figure 5A–D) along with PCR tests, and periodic acid–Schiff staining (Figure 5E,F), respectively.

We assume a rare, patient-specific pro-thrombotic immune reaction to the BHV implantation. Rare cases of such hyperactive, innate or adaptive immune responses have been previously described [34]. Specific treatment aimed at suppressing NETosis or dissolving NETs can become a promising strategy for the prevention of immunothrombosis [34]. Although there is no established anti-NET clinical therapy, a number of anti-inflammatory, antioxidant, and anticoagulant drugs, including heparin (which, however, was inefficient in this particular case), have been proposed as the possible candidates [35]. However, we could not assess the levels of the circulating NET markers (e.g., citrullinated histone H3) due to the absence of serum or plasma of this patient in our biobank; we suggest that their measurement prior to the heart valve replacement might be useful in assessing the risk of immunothrombosis.

## 4. Conclusions

We encountered a rare case of recurrent BHV and left atrial thrombosis in the early postoperative period upon a mitral valve replacement, that eventually led to the death of the patient. This case was notable for the acute neutrophil attack of the implant and formation of NETs within the clot on the BHV surface. While neutrophils are frequently observed in the blood clots, here we have demonstrated for the first time that they can also contribute to a BHV thrombosis. The current evidence suggests that neutrophils initiate immunothrombosis which is resistant to anticoagulant therapy. Suppression of NETosis or pharmacological disintegration of NETs can be among the viable treatment options for those patients with BHV thrombosis that is refractory to anticoagulant therapy. In addition, this case highlights the necessity of performing genetic testing for thrombophilia in patients planning to undergo a heart valve replacement.

## Figures and Tables

**Figure 1 ijms-23-06736-f001:**
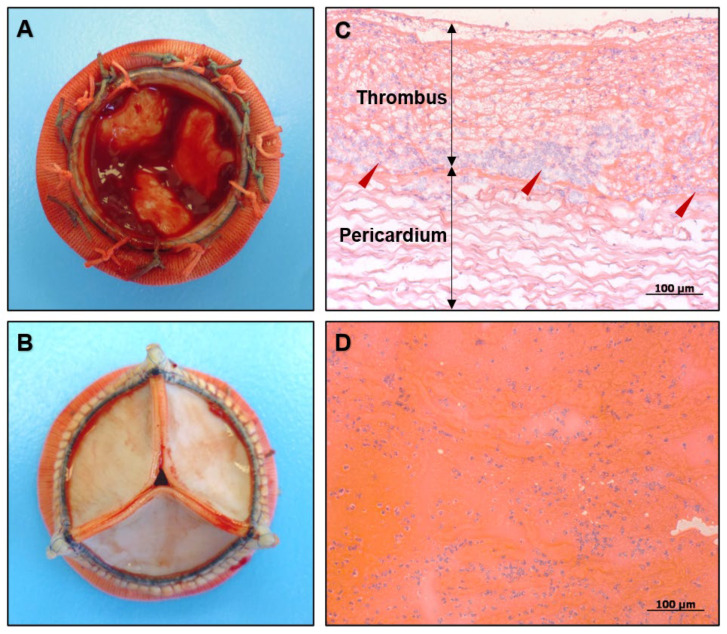
General examination of the excised bioprosthetic valve. (**A**) A massive thrombus located on the inflow side of the excised BHV; (**B**) Thrombosis-free outflow side of this BHV; (**C**) Polymorphonuclear leukocytes (indicated by red arrows) heavily infiltrated the thrombus (top) but did not invade the prosthetic leaflets (bottom). Hematoxylin and eosin staining, magnification ×200. Scale bar = 100 µm; (**D**) Left atrial thrombus also contained numerous polymorphonuclear leukocytes. Hematoxylin and eosin staining, magnification ×200. Scale bar = 100 µm.

**Figure 2 ijms-23-06736-f002:**
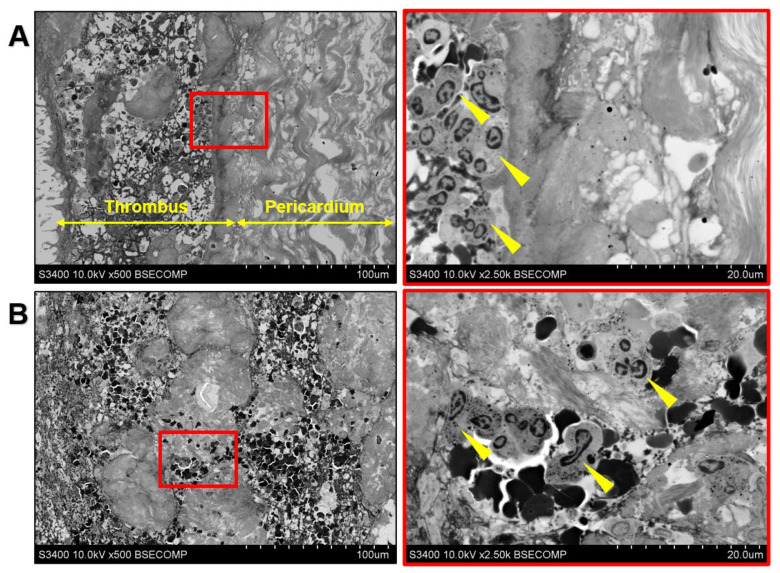
Ultrastructural analysis of thrombotic masses and prosthetic tissues. (**A**) Neutrophil-free BHV (pericardium) as compared to the neutrophil-rich clot (thrombus). Right image represents a close-up of the left image (demarcated by a red contour). Yellow arrows at the left image demarcate thrombus and bioprosthetic bovine pericardium. Yellow arrows at the right image indicate neutrophils. Magnification ×500, scale bar = 100 µm (left image) and magnification ×2500, scale bar = 20 µm (right image); (**B**) Neutrophil-rich left atrial clot. Right image represents a close-up of the left image (demarcated by a red contour). Yellow arrows at the right image indicate neutrophils. Magnification ×500, scale bar = 100 µm (left image) and magnification ×2500, scale bar = 20 µm (right image). Backscattered scanning electron microscopy (EM-BSEM method) [15], accelerating voltage 10 kV.

**Figure 3 ijms-23-06736-f003:**
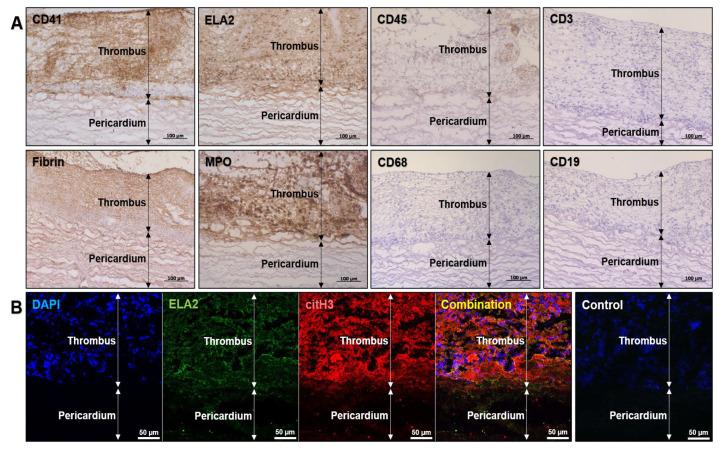
Neutrophil invasion and abundant NETs indicate immunothrombosis of BHV. (**A**) Immunohistochemical staining of the prosthetic thrombus identified platelets (CD41-positive staining), fibrin, and neutrophils (ELA2- and MPO-positive staining). CD45 staining was inconsistent. Macrophages (CD68), T cells (CD3) and B cells (CD19) were not detected. Magnification ×200. Scale bar = 100 µm; (**B**) Immunofluorescence staining showed the co-localization of citrullinated histone H3 with neutrophil elastase (ELA2) and nuclear dye (4′,6-diamidino-2-phenylindole, DAPI) within the thrombus indicating the formation of NETs. Sections without the primary antibody staining but still counterstained with DAPI were used as a negative control. Magnification ×200. Scale bar = 50 µm.

**Figure 4 ijms-23-06736-f004:**
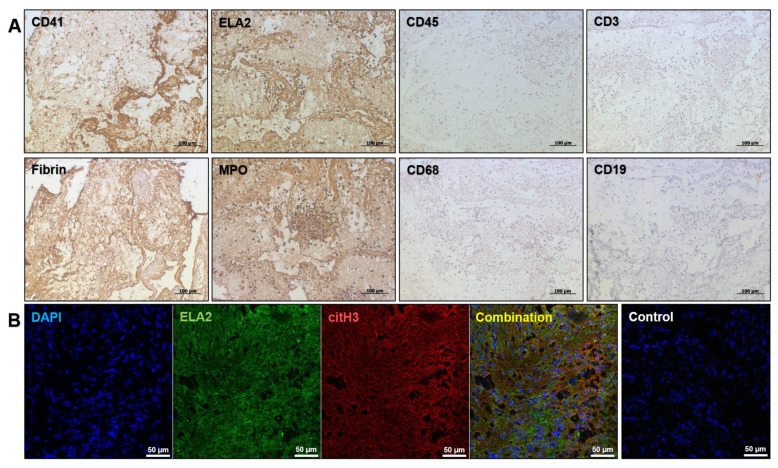
Neutrophil invasion and NETs in the left atrial thrombus confirm immunothrombosis. (**A**) Immunohistochemical staining of the left atrial thrombus identified platelets (CD41-positive staining), fibrin, and neutrophils (ELA2- and MPO-positive staining). CD45 staining, macrophages (CD68), T cells (CD3) and B cells (CD19) were not detected. Magnification ×200. Scale bar = 100 µm; (**B**) Immunofluorescence staining showed the co-localization of citrullinated histone H3 with neutrophil elastase (ELA2) and nuclear dye (4′,6-diamidino-2-phenylindole, DAPI) within the thrombus indicating the formation of NETs. Sections without the primary antibody staining but still counterstained with DAPI were used as a negative control. Magnification ×200. Scale bar = 50 µm.

**Figure 5 ijms-23-06736-f005:**
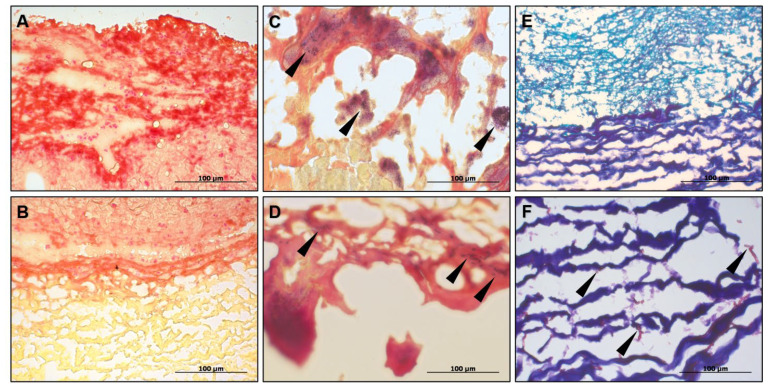
Immunothrombosis of BHV is not related to prosthetic valve endocarditis. (**A**) Negative Gram staining of the clot at the prosthetic surface. Magnification ×400. Scale bar = 100 µm; (**B**) Negative Gram staining of the clot and underlying prosthetic tissues. Magnification ×400. Scale bar = 100 µm; (**C**) Positive control for Gram staining (bacterial vegetations on the leaflet excised from the patient with prosthetic valve endocarditis). Black arrows indicate bacteria. Magnification ×400. Scale bar = 100 µm; (**D**) Positive control for Gram staining (bacteria within the microthrombi on the leaflet surface excised from the patient with prosthetic valve endocarditis). Black arrows indicate bacteria. Magnification ×400. Scale bar = 100 µm; (**E**) Negative periodic acid–Schiff staining of the clot and underlying prosthetic tissues. Magnification ×400. Scale bar = 100 µm; (**F**) Positive control for periodic acid–Schiff staining (intact bovine pericardium intentionally contaminated with the fungi). Black arrows indicate the fungi. Magnification ×400. Scale bar = 100 µm.

**Table 1 ijms-23-06736-t001:** Coagulation parameters before and after the mitral valve replacement.

Coagulation Parameters	Before inVR (22 January)	Between inVR and repVR (31 January)	After the repVR (2 February)	Before Patient Death (4 February)	Reference Range
International normalized ratio	1.20	1.25	1.33	1.18	0.9–1.5 (healthy individuals) 2.0–3.0 (those on anticoagulant therapy)
Prothrombin time (Quick-type), %	69	63	54	71	80–120
Soluble fibrin monomer complexes, mg/100 mL	17.0	6.5	4.5	10.0	<4.0
Activated partial thromboplastin time, sec	29	53	115	50	24–35
Fibrinogen, g/L	5.2	3.0	3.5	3.3	2–4
Protein C activity, %	80.0	ND	ND	ND	70–140
Antithrombin III, %	86.3	ND	ND	ND	80–120
XIIa-dependent fibrinolysis, min	14	ND	ND	ND	4–12

inVR—initial valve replacement; repVR—repeated valve replacement; ND—not defined.

**Table 2 ijms-23-06736-t002:** Complete blood count before the mitral valve replacement, between initial and repeated valve replacement, and before the fatal outcome.

Complete Blood Count Parameters	Before inVR (21 January)	Between inVR and repVR (31 January)	After the repVR (2 February)	Before Patient Death (4 February)	Ref. Range
Erythrocyte sedimentation rate, mm/h	17	24	59	51	2–15
Hemoglobin, g/L	155	114	97	98	117–145
Red blood cells, ×10^12^/L	4.9	3.6	3.3	3.4	3.7–4.7
Platelets, ×10^9^/L	253	94	69	15	170–350
Hematocrit, %	45	33	29	30	36–42
White blood cells, ×10^9^/L	10.0	14.0	8.7	10.9	4.0–8.8
Band neutrophils, %	5	2	10	7	1–6
Segmented neutrophils, %	80	92	78	84	45–72
Eosinophils, %	0	0	0	0	0–5
Basophils, %	0	0	0	0	0–1
Lymphocytes, %	9	2	7	5	18–40
Monocytes, %	6	4	5	4	3–9

inVR—initial valve replacement; repVR—repeated valve replacement; ref. range—reference range.

**Table 3 ijms-23-06736-t003:** Genetic testing for inherited thrombophilia.

Gene	Polymorphism	Genotype	Protein	Physiological Effect in the Patient Case
**Coagulation Factors**				
*F2*	rs1799963	G/G	Factor II (prothrombin)	None
*F5*	rs6025 (Leiden)	G/G	Factor V (proaccelerin)	None
	rs6027	G/G		None
*F7*	rs6046	G/G	Factor VII (proconvertin)	None
*F13A1*	rs5985	G/T	Factor XIII α chain (fibrin-stabilizing factor)	Lower fibrin clot permeability, prolonged clot lysis time, higher endogenous thrombin potential [16]
*FGB*	rs1800790	G/G	Fibrinogen β chain	None
**Platelets**				
*ITGA2*	rs1126643	C/T	Integrin α2 (CD49b)	Increased platelet aggregation [17]
*ITGB3*	rs5918	T/C	Integrin β3 (CD61)	Increased fibrinogen and platelet adhesion, accelerated clot retraction [18]
**Fibrinolysis**				
*SERPINE1*	rs1799889	4G/4G	Plasminogen activator inhibitor-1	Higher PAI-1 serum level, impaired fibrinolysis [19]

## Data Availability

The data presented in this study are available on request from the corresponding author. The data are not publicly available due to the privacy concerns of the patient’s relatives.

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
