# Peer review of "Early Postoperative Immunothrombosis of Bioprosthetic Mitral Valve and Left Atrium: A Case Report"

_ijms, 2022, doi:10.3390/ijms23126736_

Round 1

Reviewer 1 Report

The authors have presented a precious case story regarding occurred thrombosis at BHV site after replacement in a few days. The following comments are below.

1. reference number #23 has shown neutrophils bind to ECs, inappropriate describe as neutrophil infiltration in the discussion.

2. What are the biggest differences between primary thrombus by thrombectomy and thrombus on gross of fig 1. Any differences in neutrophil level if you have looked at that?  

3. What major mechanism of a massive thrombosis has been triggered by Neutrohpil infiltrated thrombus post-surgery while no big differences between INR and PT? Dose patients have been diagnosed with calcification in the coronary artery?

3. Can you provide a CBC data including levels of platelet before and after replacement surgery

4. Minor checks required such like mg% 

Author Response

We sincerely thank the reviewer for the detailed comments and constructive criticism which helped us to improve the paper. Please see the attachment.

Reviewer 2 Report

This study is a case report of a patient who experienced recurrent bioprosthetic mitral valve and left atrium thrombosis in the context of persistent atrial fibrillation, which ultimately led to death.

The thrombotic events were presented to be immune-related (immunothrombosis) due to thrombus infiltration by numerous neutrophils and the presence of NETs. No signs of infection could be detected.

The underlying cause and mechanism of this phenomenon remain unknown. If possible, it would be worthwhile to provide additional data to better understand what happened.

This case report could indeed be a good example justifying the need for better patient risk stratification and improved health care.

Detailed comments:

1. Since neutrophils seem to be incriminated, the authors should show differential blood cell counts and their evolution with time.

2. The levels of circulating NET markers could be shown (if possible). The authors should discuss about potential role of these markers in risk assessment of mitral valve disease patients prior to surgical valve replacement. 

3. Regarding the patient clinical characteristics, could antiphospholipid syndrome (or any autoimmune disorders) be excluded?

4. The authors state that heparin was inefficient, without presenting detailed anticoagulant strategy that was used peri-operatively, during surgery and at follow-up. Also, what anticoagulant was used for atrial fibrillation and when was it stopped?

5. Figure 3 shows results of immunohistochemical analyses of BHV thrombus, while the text mentions that NETs were also detected in left atrial thrombus. The later data should be shown or the text should be adapted.

6. In Figure 3A, CD45 staining shows no leukocytes in thrombi, which is inconsistent with the other data. This should be clarified or another staining protocol should be used.

7. Overall, the authors conclude on recurrent immunothrombosis (incl. in the title). However, the presence of immunothrombosis could only be demonstrated on explanted BHV obtained after the first surgery. The conclusions are therefore not correct. The title should be modified as "recurrent thrombosis" or the term "recurrent" should be deleted.        

Author Response

(The authors gave the same response as above.)

Reviewer 3 Report

Authors described an interesting case of a 72-year-old female patient with mixed rheumatic mitral valve disease and persistent atrial fibrillation who underwent mitral valve replacement and suffered from a combined thrombosis of the bioprosthetic valve and the left atrium as soon as 2 days postoperation. Authors performed also a detailed histopathological analyses supported by nice images.

  1. Was any diagnostic towards thrombophilia conducted? I understand that the patient died, but many investigations were performed, bo maybe also blood for testing in that direction was saved?
  2. I would elaborate a bit more on thrombus management in an acute setting? Did authors consider any GPIIb/IIIa or fibrinolysis?

Author Response

(The authors gave the same response as above.)

Round 2

Reviewer 2 Report

The manuscript was substantially improved. All reviewer's comments have been adequately addressed.